# Integrating EEG and Machine Learning to Analyze Brain Changes during the Rehabilitation of Broca’s Aphasia

**DOI:** 10.3390/s24020329

**Published:** 2024-01-05

**Authors:** Vanesa Močilnik, Veronika Rutar Gorišek, Jakob Sajovic, Janja Pretnar Oblak, Gorazd Drevenšek, Peter Rogelj

**Affiliations:** 1Faculty of Medicine, University of Ljubljana, 1000 Ljubljana, Sloveniajanja.pretnar@kclj.si (J.P.O.); gorazd.drevensek@mf.uni-lj.si (G.D.); 2University Medical Centre Ljubljana, 1000 Ljubljana, Slovenia; veronika.rutargorisek@kclj.si; 3Faculty of Mathematics, Natural Sciences and Information Technologies, University of Primorska, 6000 Koper, Slovenia; peter.rogelj@upr.si

**Keywords:** EEG, functional connectivity, neural network classification, Broca’s aphasia

## Abstract

The fusion of electroencephalography (EEG) with machine learning is transforming rehabilitation. Our study introduces a neural network model proficient in distinguishing pre- and post-rehabilitation states in patients with Broca’s aphasia, based on brain connectivity metrics derived from EEG recordings during verbal and spatial working memory tasks. The Granger causality (GC), phase-locking value (PLV), weighted phase-lag index (wPLI), mutual information (MI), and complex Pearson correlation coefficient (CPCC) across the delta, theta, and low- and high-gamma bands were used (excluding GC, which spanned the entire frequency spectrum). Across eight participants, employing leave-one-out validation for each, we evaluated the intersubject prediction accuracy across all connectivity methods and frequency bands. GC, MI theta, and PLV low-gamma emerged as the top performers, achieving 89.4%, 85.8%, and 82.7% accuracy in classifying verbal working memory task data. Intriguingly, measures designed to eliminate volume conduction exhibited the poorest performance in predicting rehabilitation-induced brain changes. This observation, coupled with variations in model performance across frequency bands, implies that different connectivity measures capture distinct brain processes involved in rehabilitation. The results of this paper contribute to current knowledge by presenting a clear strategy of utilizing limited data to achieve valid and meaningful results of machine learning on post-stroke rehabilitation EEG data, and they show that the differences in classification accuracy likely reflect distinct brain processes underlying rehabilitation after stroke.

## 1. Introduction

A stroke is a profound, life-altering occurrence affecting around 12 million individuals globally, encompassing both ischemic and hemorrhagic variants [1]. The mortality rate among those affected ranges from 20% to 50%, contingent on the stroke type and the accessibility of high-quality care [1,2,3]. Moreover, the significant majority of survivors are faced with substantial disability or dependency as a consequence [1]. A distinctive set of post-stroke symptoms is encapsulated in Broca’s aphasia—a condition marked by impaired fluency in speech due to damage to Broca’s area, responsible for speech production [4]. Since speech is a fundamental aspect of human identity and essential for various daily activities involving communication, there arises a critical need for the development of effective rehabilitation methods for individuals with post-stroke Broca’s aphasia. Investigating the mechanisms underlying the damage causing aphasia and those facilitating recovery emerges as a pivotal area of research [5].

The process of recovery is a complex interplay of various factors and processes that the brain undergoes following a stroke [6,7]. Although inextricably linked, it can be divided into two broader processes. The first is the structural change of the brain following a stroke, the extent of which depends on the type of stroke, the site of the stroke, its extent, and other clinical factors, such as the time to treatment and the treatment itself. Invariably, parts of the tissue die, resulting both in white matter loss, representing the loss of communication highways, and gray matter loss, representing the loss of activity-producing nodes in the brain [7,8,9,10]. Structural alterations are thus reflected in both the impaired production of brain activity and the transmission of produced activity to other parts of the brain. The process of rehabilitation and healing is associated with the structural changes of the brain via the generation of new white matter tracts, representing the establishment of new connections, and by the thickening and strengthening of previously existing, but weak connective fibers [7,8,9,10,11,12]. These structural changes are reflected in the second process, the functional reorganization of the brain. Usually, this includes the recruitment of contralesional areas of the brain to perform the functions of the damaged side; the recruitment of adjacent, functionally related areas to perform some of the functions of the damaged area; and the extensive reorganization of the information flow between functionally distinct areas of the brain, to compensate for impaired task- or specific function-related processing [9,13,14]. In terms of stroke to the frontal regions of the brain, resulting in difficulties with speech production, first, the entire language-related left fronto-temporal network, the general-processing bilateral frontal network, and the contralesional equivalent to the affected site are recruited to take over the function of the damaged area in the short and medium term. Gradually, the areas adjacent to the site of the stroke form their own network nodes and integrate into the reorganized interplay of networks that have, up to this point, compensated for the damage sustained by the stroke [4,7,15].

In recent years, the use of electroencephalography to predict outcomes after stroke and to enhance rehabilitation has gained attention in the published literature [16]. Connectivity measures are often extracted from electroencephalography (EEG) data and used as inputs for machine learning algorithms, as they provide detailed insights into the reorganizing processes of the brain. Several reviews and meta-analyses have shown that EEG has good predictive value for several functions that can be impaired after stroke, from motor function to speech and cognition. Machine learning is increasingly being applied in the field of stroke rehabilitation, particularly as a tool to personalize therapy and monitor progress [16,17,18,19]. This is especially relevant given the wide variability in stroke symptoms and recovery trajectories. Machine learning algorithms can analyze large amounts of patient data, including neuroimaging, motor performance metrics, and other clinical indicators, to predict recovery outcomes and customize treatment plans. Input features reflecting brain connectivity, derived from EEG data, often include the phase-locking value (PLV), weighted phase-lag index (wPLI), and Granger causality (GC). The first two measures are undirected and reflect the phase synchronization of brain activity, while GC is a directed (effective) measure, based on the predictive capacity of the activity of one electrode on another [13,20,21].

The predictive accuracy of recovery success, using various features of EEG signals, usually ranges from 70 to 90%, depending on the function for which the outcome of recovery is being predicted and the design of the study attempting to predict it. The main advantages of using both machine learning to predict and EEG to assist in the process of rehabilitation are the reduction of bias in the process of rehabilitation planning, the diagnosis of aphasia itself, and the recognition of progress [16]. There are few studies, however, that have focused on distinguishing the brain activity of people before and after rehabilitation, representing a gap in the current research that the present study aims to fill. More importantly, developing machine learning models that are able to distinguish between pre- and post-rehabilitation brain activity can provide novel, crucial insights into the process of rehabilitation, by backtracking the classification process and making it explainable. Thus, the present study aims to contribute to the current body of knowledge by combining machine learning with a rigorous testing approach, the use of careful and sophisticated connectivity analysis of EEG signals, and a focus on the effects of rehabilitation on the brain.

### 1.1. Related Works

To the best of our knowledge, very few studies that have utilized modern machine learning approaches have used EEG data-derived features to either predict the recovery of patients with aphasia or to classify their recovery status. There seem to be two available studies that tackle this research question. Clerq et al. [22] used a support vector machine to detect aphasia in EEG recordings of healthy subjects and individuals with aphasia. The EEG was recorded while the subjects listened to a 25-min story, and mutual information (MI) connectivity measures in the delta, theta, alpha, beta, and gamma bands were extracted. The authors were able to obtain 88% accuracy, using the data of 27 participants. On the other hand, Chiarelli et al. [23] reported predictive capacity with r = 0.53 and AUC = 0.8 for their non-linear support vector regressor, predicting the functional recovery of 101 patients with monohemispheric stroke. They reported that the usage of delta, theta, alpha, and beta power data alongside the NIH stroke scale improved the prognostic capacity when compared to the usage of the stroke scale alone.

More broadly, a clinical focus article and a systematic review on the topic of the prediction of aphasia rehabilitation using neuroimaging and EEG reveal three further studies that can be compared to this one [24,25]. Although none of these used modern machine learning approaches on EEG data, their results can still be informative for the current study. The first study by Iyer et al. [26] used dynamic causal modeling (a method of directed connectivity estimation) on source-reconstructed, high-density EEG recordings during a semantic and phonological picture–word judgment task, to predict language recovery in ten patients with chronic aphasia. Patients underwent a four-week language rehabilitation program, before and after which the event-related potentials of the two tasks were captured with EEG. Using dynamic causal modeling, the authors identified three broader systems of connections that can be associated with language recovery. These are the connection between the left inferior parietal lobule (IPL) and the left inferior frontal gyrus (IFG), the connection between the right IPL and the right anterior medial temporal gyrus (AMTG), and the connection between the left and right IPLs. The authors report an r of 0.63 for the left IPL to IFG connection, −0.76 for the right IPL and right AMTG, and r = 0.77 for the left to right IPL connection. This reveals the utility of effective connectivity approaches to identify brain systems crucial for language recovery.

The next study is the one by Nicolo et al. [27], who used network measures of the coherence of resting EEG signals in the delta, theta, alpha, and beta bands, to explain the motor and language recovery of 24 patients in the period between 2–3 weeks after stroke and 3 months after stroke. The authors then validated their results with 18 more patients. They found that for language recovery, a greater weighted node degree 2–3 weeks after stroke over Broca’s area in both the beta and theta bands was associated with better recovery (r = 0.7 for both). Furthermore, an increase in node degree from the first to the second recording was conversely associated with poorer recovery (both beta and theta bands, r = −0.6 and r = −0.8, respectively). These results showed that activity coherence measured over Broca’s area is crucial for the beginning stages of recovery, but might be detrimental as more time passes after the stroke. Moreover, the authors found that connections between the affected and the contralateral area were also crucial for the recovery process (r = 0.5).

The third study, by Szelies et al. [28], used the laterality indices of the delta, theta, alpha, and beta bands of the resting EEG of 23 patients with aphasia, 2 weeks after stroke. The laterality index was used to determine the relative left accentuation of EEG power in a given band. Using a stepwise discriminant analysis of theta and alpha laterality (but not other bands), they successfully discriminated between relatively good and poor outcomes of rehabilitation (r = −0.88 for alpha and r = 0.63 for theta). A combination of both gave the best result (r not reported). These results once again confirm the importance of theta power over the affected hemisphere in predicting recovery, as well as showing that the same pattern of alpha activity might be associated with poorer recovery.

### 1.2. Contributions and Article Organization

In comparison with the above-listed previous studies, this paper offers a unique contribution in the form of an advanced EEG analysis, utilizing functional and effective connectivity metrics, comparing their utility when used as input features for machine learning and taking advantage of modern machine learning approaches to specifically target the rehabilitation of Broca’s aphasia following a stroke. The goal of this study was the creation of a simple machine learning model that could distinguish between pre- and post-rehabilitation EEG data, based on neuroscientifically relevant features. Thus, it was designed as a feasibility study of this approach, where the model’s predictive value and accuracy were thoroughly tested, and the process of generating a machine learning model was also thoroughly evaluated. Moreover, the various possible input features were also evaluated for their predictive capacity. This study focused on developing an accurate machine learning model of brain activity during a verbal working memory task, which was able to recognize the distinguishing characteristics of the brain activity of stroke patients with Broca’s aphasia before and after rehabilitation. Additionally, to provide a contrast with a language-oriented task, the model was also trained on the data obtained during a spatial working memory task and the accuracies were compared.

Our paper is organized as follows. Materials and methods: here, we explain how the EEG data were obtained, how they were pre-processed, how the connectivity measures were calculated, and how the features of the data were selected as inputs for our model. We end this section by describing our machine learning model in detail and explain the process of its testing and validation. Results: we begin this section by providing descriptive data on the accuracies of our machine learning model, separated by the measure used as its input during training and the task during which the data were recorded. We continue by providing a visualization of the accuracy of our model, averaged over the eight participants and separated by the frequency band that was used to calculate the connectivity. We provide separate figures for the two different tasks that the patients underwent, while their brain activity was captured. We end the Results section by presenting the results of an in-depth statistical analysis of the obtained accuracies, to determine which measure was the best suited as an input for our machine learning algorithm and to assess which type of task produced data that could best be used to discriminate between the state before a patient undergoes rehabilitation for stroke and after this has been carried out. Discussion: in this section, we first discuss the changes in connectivity observed due to the process of rehabilitation and continue by assessing whether we were successful in reaching the set goals of this study. We continue by connecting our results with those of previous studies informative to the topic of this paper, and we end the section by interpreting all unexpected results obtained. Conclusions: here, we present a final summary of this paper and offer some concluding remarks on the results and their meaning for future research.

## 2. Materials and Methods

### 2.1. EEG Data Acquisition

To investigate the diverse machine learning techniques and their efficacy in classifying data as either pre- or post-rehabilitation, we utilized the electroencephalogram (EEG) data from eight patients with Broca’s aphasia resulting from a stroke. These patients underwent verbal and visuospatial working memory tasks both prior to and following their rehabilitation, as outlined by Sternberg [29] in Rutar Gorišek et al. [21]. The EEG tests and recordings were, on average, conducted 54.4 (±SD 30.7) days following the occurrence of the ischemic stroke [21].

The EEG was recorded from a 128-channel device, with the electrodes mounted in an elastic cap with the 5–5 standard positioning system. The recording hardware consisted of BrainAmp amplifiers (Brain Products GmBH, Gilching, Germany). A reference-free montage was used, with the referencing being done via the common average reference approach. The sampling rate was 500 Hz and the impedance of individual electrodes was kept below 5 kΩ. Of the 128 electrodes mounted, 6 were used to record electrooculogram data, leaving 122 EEG data channels. The BrainProducts recorder software was used to record and digitally store the data (Brain Products GmBH, Gilching, Germany).

The 10 patients were selected based on the Boston Diagnostic Aphasia Evaluation (BDAE) and right-handedness based on Edinburgh Handedness Inventory (EHI), their age ranged from 47 to 85, and they were, on average, 67 years of age [21]. Additional inclusion criteria were that the stroke that the patient experienced was their first one, it occurred in the territory of the M2 segments of the left middle cerebral artery, and an observable lesion was present in Broca’s area or its vicinity, or in part of Broca’s complex. The exclusion criteria were the presence of a known psychiatric or neurological disorder prior to stroke, moderate to severe hemiparesis, severe complications or worsening of the patient’s condition, more than a single stroke, or the presence of other intracranial pathologies (e.g., tumor or vascular leucopathy). Figure 1 shows the density map of the lesions, depicting where these were most commonly observed. MRIcron v. 8/2014 was used to create the image, from diagnostic MRI images.

All patients participated in two working memory tasks based on the Sternberg task [29]—a verbal working memory task and a visuospatial working memory task. The EEG recordings were taken before and after rehabilitation, while the patients were performing 80 trials of the same task on a computer. One trial consisted of the following: item presentation (encoding the letters’ sequence or position depending on the type of task), a blank screen (maintenance of the presented item in one’s working memory that lasted for 4 s), a question, a mouse-click response (motor response), and a blank screen before the next trial (a 4-s resting period) [21]. Every event was separately marked in the patients’ EEG signal, and we selected the maintenance part of every trial. The present study used the data from 8 patients, as two were missing data for one of the conditions. Otherwise, the use of the data from only 8 patients was a limitation, but it was mitigated by the rigorous leave-one-out cross-validation approach, which maximized the use of the available data and enhanced the generalizability of the model.

### 2.2. EEG Data Pre-Processing

We decided against using the raw EEG data as the machine learning input. There were three major reasons for this. The first is that we were interested in classifying actual brain activity, not a change in artifacts. For example, if a patient accidentally (or due to the recovery process itself) blinks less after rehabilitation, the machine learning model would focus more on the eye-blink artifact detection and classification, not on the change in brain activity. The second reason was that our sample size was limited, which made artifact suppression through sample size (as the artifacts should be largely randomly occurring over subjects) unfeasible in this study. Furthermore, providing the informative features as an input to the classifier reduces its required complexity and increases the trainability with a smaller amount of input data and smaller number of patients included in the study. The third reason is that we wished to enable the extension of this work towards explainability that could contribute to the development of neuroscience. The estimated importance of specific features is, from this perspective, more beneficial than a huge amount of parameters for an artificial neural network.

We employed the following pre-processing protocol.

Visual inspection of data. Visual inspection was carried out, to detect any large deviations from the expected EEG signal, e.g., malfunction of the apparatus, large drift of the signal, heavy artifacts below 0.1 Hz. If a dataset exhibited such characteristics, it had to be removed from further analysis; however, no datasets were removed at this step.Filtering of the data. First, a band-stop filter was applied, to remove line noise. The filter was a Hamming-windowed sinc finite impulse response (FIR) one-pass-zero-phase filter, with the order of 990, a cutoff between 48 Hz and 52 Hz, a transition width of 2.0 Hz, a maximum pass-band deviation of 0.22%, and stop-band attenuation of −53 dB. A one-pass-zero-phase Hamming-windowed sinc FIR band-pass filter was also applied, between 1 and 70 Hz and with a 1 Hz transition band; the order of the filter was 1650, and the pass-band was between 1.5 and 69.5 Hz, with a maximum pass-band deviation of 0.22%. The stop-band attenuation was −53 dB.Epoching of the data. We extracted data epochs for the two tasks performed during data measurement, i.e., the verbal memory task and the spatial memory task. The time locking events were stimulus presentations for each task. The epoch length was 1100 ms, with 100 ms prior to event presentation being included as a baseline.Automatic rejection of channels with bad data. Channels with normalized activity above or below 5 SD with regard to all other channels were removed from the data.Independent component analysis (ICA) decomposition of the data. We carried out the decomposition of data using the infomax ICA algorithm of Bell and Sejnowski [30] with the natural gradient feature of Amari, Cichocki, and Yang [31] and the extended ICA algorithm of Lee, Girolami, and Sejnowski [32].ICA component removal. We removed any component that represented artifacts, such as eye blinks, swallowing, lateral eye movements, and heavy muscle artifacts.Baseline removal. The epoch baseline was removed at this point, with the 100 ms before stimulus presentation serving as the baseline.Automatic epoch rejection. Any epoch exceeding ±50 μV at any point in the epoch was removed from the data.Re-referencing to average and interpolation of missing channels. Spherical interpolation following the procedures in Feree was carried out [33].

After we pre-processed the data, further data preparation was carried out. Namely, we calculated several connectivity measures that were then used as feature inputs to the machine learning model. We selected only the most statistically significant connectivity features and used them to train the neural network classification model.

### 2.3. Functional Connectivity Calculation

For each patient and each individual epoch, we computed functional connectivity features using several different connectivity estimation methods. For the verbal working memory task, we obtained 582 epochs of clean data from the recordings of patients before rehabilitation, and 577 epochs after rehabilitation in total. For the visuospatial working memory task, there were 581 epochs before rehabilitation and 669 epochs after. Every combination of connectivity method and frequency range was calculated for every epoch. Functional connectivity methods can be divided into three categories, with the first one including the methods based on phase lag: PLV, wPLI, and complex Pearson correlation coefficient (CPCC) [34,35]:(1)$$PLV\left(x_{1},x_{2}\right)=\left|\frac{1}{N}\sum_{n=1}^{N}e^{i(\phi_{x_{1,n}}-\phi_{x_{2,n}})}\right|,$$
(2)$$wPLI\left(x_{1},x_{2}\right)=\frac{|\sum_{n=1}^{N}Im\left(x_{1,n}\cdot x_{2,n}^{*}\right)|}{\sum_{n=1}^{N}\left|Im\left(x_{1,n}\cdot x_{2,n}^{*}\right)\right|},$$
(3)$$CPCC\left(x_{1},x_{2}\right)=\frac{\sum_{n=1}^{N}x_{1,n}\cdot x_{2,n}^{*}}{\sqrt{\sum_{n=1}^{N}{\left|x_{1,n}\right|}^{2}}\cdot \sqrt{\sum_{n=1}^{N}{\left|x_{2,n}\right|}^{2}}}.$$Here, $x_{1}$ and $x_{2}$, are complex number analytic signals obtained from the real electrode signals $x_{1}$ and $x_{2}$ using the Hilbert transform, *N* is the number of samples, and ${\left\{.\right\}}^{*}$ is the complex conjugate operator. PLV quantifies the consistency of the phase difference between two signals. If the phase difference tends to remain constant over time, the PLV value will be high, indicating strong phase locking or synchronization. wPLI is designed to address the problem of the high sensitivity of PLV to volume conduction. It also measures the consistency of the phase difference, but is limited to its imaginary component, which cannot be caused by volume conduction. CPCC yields complex-valued results [35], where the absolute value absCPCC reflects the total connectivity, while imaginary component iCPCC reflects only the part that cannot be subjected to volume conduction effects. We also investigated the use of both components, i.e., real and imaginary, as CPCC.

The second group was information-based with the MI connectivity estimation measure [34]. This is a measure from information theory that quantifies the statistical dependence of two signals from their instantaneous amplitudes:(4)$$MI\left(x_{1},x_{2}\right)=H\left(x\right)+H\left(y\right)-H\left(x,y\right)$$Here, H(.) and H(.,.) stand for signal entropy and joint entropy, respectively.

In contrast to the first two groups of measures that estimated undirected connectivity, the third was a directed one, a prediction model-based analysis known as GC [34]. In the context of electrode signals, GC examines the ability of one source electrode’s past values to predict the future values of a target electrode. The rationale is that the inclusion of the source electrode’s past information in the prediction model should reduce the prediction error for the target electrode if there is a directed influence:(5)$$GC\left(x_{1},x_{2}\right)=log\left(\frac{Var(e_{1})}{Var(e_{1,2})}\right)$$Here, $e_{1}$ and $e_{1,2}$ are the signal prediction errors of univariate and bivariate vector autoregressive models, respectively.

All measures based on phase lag can be used only with signals with a narrow frequency band, which is a limitation of the Hilbert transform [34] used to estimate the instantaneous signal phase angles. Similarly, the frequency band must also be limited for MI, in order to avoid mixing the estimated signal-level relationships of underlying neural processes reflected at different frequency bands, which would veil the actual signal interdependence. Consequently, we divided the EEG data into the following frequency bands: delta (0.5–4 Hz), theta (4–7 Hz), low gamma (30–45 Hz), and high gamma (45–60 Hz). Then, we estimated the functional connectivity for each of the bands independently. Note that there is no such limitation for GC, which was calculated across the entire frequency spectrum that was left after our EEG data pre-processing, i.e., 1–70 Hz. The frequency bands were chosen as they have been shown to be associated with the symptoms and recovery of Broca’s aphasia. Moreover, they are less prone to being affected by the task itself, namely interpersonal variability in the capacity to perform the task and the variability of recovery, as both alpha and beta are strongly associated with working memory. They are also less consistently associated with language. Therefore, we decided to omit them from the analysis to reduce the number of experiments to carry out and to use the features most relevant to language recovery.

The amount of feature types along with two different tasks and two different EEG recordings was particularly large. To explain in detail, we had 8 patients that each had 2 EEG recordings, before and after rehabilitation, which we wanted to accurately classify. The patients went through 2 different tasks, and we used the data from both. There were 5 different connectivity methods used, GC, wPLI, PLV, MI, and CPCC. We separately calculated the functional connectivity with all connectivity methods, except for GC, by using 4 different frequency ranges, i.e., theta, delta, lower gamma, and higher gamma. GC used the EEG data with no specific narrower frequency band because of the nature of this method, so 1 calculation was done with GC. The number of electrode pairs used was 122, meaning that a 122 × 122 matrix was computed for each epoch of every patient and for all the connectivity estimation methods and frequency bands. All functional connectivity methods, except for GC, are non-directed methods, meaning that one half of the connectivity matrix was used (leaving us with 7442 connectivity estimates) in order to not use the same channel pairs twice. Consequently, the number of features for GC is twice as large (14,884 estimates) compared to other connectivity methods.

We decided to use these measures, as opposed to raw, continuous data, for several reasons. First, connectivity measures have been shown to describe the process of rehabilitation well and are thus salient features for machine learning. Second, the use of discrete inputs simplifies the model architecture that can be used with the data and thus enhances the potential for explainability. Third, the chosen measures reflect different processes that underlie the brain activity reflected in the EEG signal, which makes their comparison more informative than using raw data.

### 2.4. Statistical Feature Selection

To reduce the feature space for the classification of data as belonging to the pre- or post-rehabilitation phase (the two classes in our data, regardless of the connectivity method features used as input), and to help the model to generalize the knowledge and avoid overfitting, we performed a repeated-measures *t*-test for each of the method–frequency band combinations. *p*-values were calculated and only electrode pairs corresponding to the top 10% of the lowest *p*-values (i.e., 1488 features for the GC and 744 features for all other connectivity measures) were kept for use as input features for our machine learning model. This approach reduced the number of features and also provided information on which features changed the most (and most consistently) prior to and after rehabilitation. A similar approach was previously used by [36], to determine features for text categorization.

In order to better understand the functional connectivity of different methods and frequencies, we provide a few examples that showcase the top 10% of features or functional connectivity values among the electrodes that had the lowest *p*-values for a certain combination. See Figure 2 for reference.

Please note that the goal of the present study was not to provide a detailed statistical breakdown of changes in brain connectivity during rehabilitation, but instead to verify how the choice of input feature and the type of task (more or less related to the primary impairment after stroke) affects the classifier accuracy in EEG data.

### 2.5. Our Machine Learning Model

Our machine learning model was constructed using the MATLAB Deep Learning Toolbox (The MathWorks, Natick, MA, USA), in MATLAB version 2023a. The model was designed as a three-layer fully connected neural network; see Figure 3. The first layer, the input layer, allows feeding in the statistically selected top 10% of a selected feature type, extracted using a specific connectivity measure in a specific frequency band. The three fully connected layers had 10 nodes, 5 nodes, and 2 nodes, respectively, and the Leaky Rectified Linear Unit (leaky RELU) was selected as the activation function in between the fully connected layers. The softmax layer that follows is used to estimate the likelihoods of each of the classes, i.e., one for pre-rehabilitation (before) and one for the post-rehabilitation stage (after). Finally, the class output layer makes the final classification to one of the two classes (i.e., either before or after rehabilitation). The goal for our classifier was to correctly classify each epoch of data into either the pre- or post-rehabilitation class.

The data were split into a training set and a testing set in a leave-one-out cross-validation manner. Thus, we set aside all of one patient’s data epochs for testing, and the rest of the patients’ data were used for training. This was repeated eight times, which was the number of all patients, meaning that every patient’s data were at some point used to test the model’s prediction ability. The reason for choosing this methodology, instead of the type of standard division of data to 80:20 or 70:30 for testing and training sets, respectively, was to avoid the effects of the interpersonal variability of the data, which would cause the model to overfit to the data of our specific patients but be inefficient when classifying novel data. With the rotation of the patients’ data, we could normalize this by averaging the accuracies of all patients at the end, providing a clearer picture of the true accuracy of our model, regardless of whose data are used as its input—making the model as subject-invariant as possible with the dataset available.

Training options for the machine learning model were defined with iterative testing of the parameters to maximize the accuracy and the stability of predictions. We used the stochastic gradient descent with momentum (SGDM) optimizer and set the maximum number of epochs to 40, with the mini batch size of ten. The data were shuffled on every epoch. The training and validation of the model were done separately for every type of connectivity–frequency–task combination.

## 3. Results

### 3.1. Descriptive Data of Model Accuracy by Task and Connectivity Measure

Table 1 presents some basic, descriptive data on the accuracies achieved by our machine learning model, with regard to input features based on different connectivity measures.

In the analysis of EEG data pertaining to both verbal and spatial working memory, the accuracy of the predictions made by our machine learning algorithm was evaluated for each of them separately and for each combination of frequency band and functional connectivity measure. As previously highlighted, due to the pronounced inter-patient variability, we used a leave-one-out validation approach, testing the accuracy on data segments of the EEG of each patient separately.

The machine learning algorithm exhibited the best performance on connectivity features derived from the GC or PLV methods when applied to verbal working memory EEG data. Specifically, the PLV approach yielded the highest average accuracy, exceeding 80% for the lower gamma frequency band (30–45 Hz), with peak accuracies of 100% observed for both lower (30–45 Hz) and upper gamma (45–60 Hz) ranges. The wPLI demonstrated suboptimal performance across all combinations of verbal working memory data, as shown in the figure below. MI and CPCC reached accuracies of nearly 100% in all frequency bands; however, it is noteworthy that they both also recorded minimum accuracies below 50%, indicating substantial variability in their performance. MI in the theta band was an exception, showing good median and mean performance, while avoiding overfitting.

Figure 4 and Figure 5 show the accuracy of the classification of brain connectivity for the verbal memory task and the spatial memory task for all measures and frequency bands. By comparing both figures, we can observe the effect of the task, with the model accuracies fluctuating less for the verbal working memory task than for the spatial working memory task.

### 3.2. Evaluation of ML Models for the Best-Performing Methods

The efficacy of our machine learning models can be depicted through receiver operating characteristic (ROC) curves and the standard evaluation metrics for machine learning models. Precision, sensitivity, specificity, accuracy, and the F-measure were utilized for this analysis. Please note that all tests have been repeated for the sake of evaluating them; therefore, the accuracies might differ slightly from the ones in Table 1.

The ROC curves for the best-performing methods present a compelling visual indication of the models’ capabilities. If we look at Figure 6, the averaged curves are far removed from the line of no discrimination, affirming the models’ proficiency in distinguishing between the different output classes. Notably, the models applied to verbal working memory data exhibit curves that are closer to the desired top-left corner of the ROC space, signifying a superior true positive rate and a minimized false positive rate. This can be seen in the confusion matrices in Figure 7 as well.

The evaluation metrics, as presented in Table 2, provide a more granular view of the models’ performance. High precision across the models indicates a low incidence of false positives, which is crucial for applications where the cost of a false alarm is high. Sensitivity, or the true positive rate, is generally high but shows some variance among the methods. For instance, the MI theta method in the verbal working memory data and the PLV low gamma method in the spatial working memory data exhibit lower sensitivity, which could signal a tendency to miss true positives.

Specificity scores are quite high for all methods, underscoring the models’ success in correctly identifying negatives. The F-measure, which balances precision and sensitivity, is notable for its high values, especially in the GC method for the verbal working memory data and the MI theta for the spatial working memory data. This suggests not only precision in the classification but also a commendable recall rate, ensuring that the majority of relevant instances are captured by the model.

In summary, the evaluated machine learning models exhibit a substantial capacity for accurate EEG data classification. The combination of ROC curve insights and quantitative evaluation metrics provides a comprehensive overview of the models’ strengths, highlighting their potential for enhancing EEG-based analytical applications.

### 3.3. Detailed Method Comparison

The data on the accuracy of ML using individual connectivity measures were analyzed with a two-way mixed effects ANOVA, comparing all 25 feature types of connectivity measure and frequency band sets in both conditions. The averaged accuracy of all segments of data belonging to the EEG of one participant was treated as a single data point, giving us a sample size of eight. The data fit some of the assumptions for the ANOVA (equality of error variances tested by Levene’s test, sphericity), but violated the normality of distribution assumption in the case of many of the variables. Thus, a two-way Friedman ANOVA was used to account for these shortcomings of the data and verify the parametric results. All results were corrected for multiple comparisons using the Bonferroni correction. The results of the Friedman ANOVA confirmed the results of the parametric one; thus, parametric results are reported and displayed for ease of interpretation. Post-hoc test results (*t*-tests) were also corroborated by non-parametric measures (Wilcoxon signed-ranks test).

To verify whether the observed differences among the methods, frequency bands, and tasks are consistent enough to make conclusions in favor of any one of them for the prediction of rehabilitation success after Broca’s aphasia, we extensively statistically tested the obtained accuracies. Table 3 shows the results of the omnibus two-way ANOVA, where the effect of the connectivity measures chosen on the accuracy of the ML model was significant, but no difference in accuracy due to the task was detected; neither was a significant interaction between the task and connectivity measure chosen observed. The results in this table represent two different effects that we tested for, i.e., the effect of the chosen connectivity measure (and frequency band for all measures but GC) on ML model accuracy and the effect of the task during which the data were recorded (verbal or spatial working memory). The first is reflected in the row titled “Connectivity measure” and the second in the row titled “Task”. The row titled “Interaction” represents the test of the interaction between effects, effectively testing whether the accuracy of our ML model changed differently when using the data of the verbal working memory task and the data of the spatial working memory task, when switching between connectivity measures. In other words, we assessed whether the difference in accuracy between the two tasks was dependent on the connectivity measure used.

The results of the post-hoc testing for the ANOVA are shown in Table 4. The table includes all statistically significant method pairs, while conclusions regarding method pairs that are not provided in the table cannot be made, indicating similar method accuracies. The results reveal that GC and PLV outperformed many (but not all) other measures. Consequently, these two measures have the highest differences in accuracy with regard to other measures, between 23% and 39%. The worst performers are the wPLI and iCPCC measures.

## 4. Discussion

EEG is, in conjunction with machine learning, transforming the field of rehabilitation. This potent combination facilitates a deeper understanding of brain activity, offers more precise predictions for recovery outcomes, and enables the personalization of treatment plans [37]. Brain connectivity analysis can, by itself, contribute to the understanding of neural processes. Figure 2 shows the top 10% of connections that change due to the rehabilitation process, as obtained by GC. For the GC, we see a reorganization of connectivity, with the electrodes on the right side of the head (contralateral to the site of stroke) exhibiting a larger amount of connections, both with their neighbors and with more distant recording sites. Additionally, we see a strengthening of connectivity from the left parieto-occipital recording sites, as well as the sites situated over the left fronto-temporal areas with near and distant electrode sites.

In this study, our main aim was to create a precise machine learning model that can determine the difference in the condition of patients with Broca’s aphasia before and after rehabilitation. It has already been shown that brain connectivity is a reliable indicator of how the brain activity reorganizes after a stroke and subsequent rehabilitation [38]. We focused on analyzing brain connectivity during tasks involving verbal and spatial working memory. Despite having a small dataset with information from only eight subjects, we successfully developed a neural network-based classifier model. We opted for simplicity to avoid potential issues like overfitting. The results obtained using the best-performing connectivity measure show good consistency and 89.4% mean/95% median accuracy. For all results, see Table 3.

When compared to the five related works that also use EEG data to either predict recovery from [23,26,27,28] or to detect aphasia [22], the present study explores more than one connectivity measure, while also providing a contrast to the EEG connectivity during a language-related working memory task with the connectivity during a non-language-related task. Moreover, while [22] reports being able to distinguish between individuals with aphasia and healthy controls, our model distinguishes patients based on pre- and post-rehabilitation data, modeling the recovery process instead of the difference between healthy individuals and patients with stroke. When compared to [23], we do not predict the extent of recovery, and we use a discrete classifier, while theirs was a regression model. Moreover, more complex measures were used in the present paper, extracting more detailed information about the recovery process. Similarly to the study in [26], we also note the strengthening of parietal and frontotemporal electrode connections during the process of rehabilitation, when examining the GC connectivity in Figure 2. The previous study, however, also used a regression model to describe to what extent the rehabilitation was successful, not only whether it was present or not. In contrast, we use varied connectivity measures, possibly capturing several brain processes underlying rehabilitation, although on the level of the scalp, not in the source-reconstructed space, as in the work of Iyer et al. [26]. Looking at the studies of [27,28], we also find that the theta band can be an important predictor of rehabilitation, but the differences in our study designs and the analysis of EEG data preclude us from drawing further parallels between the results.

The accuracy analysis revealed that the machine learning algorithm showed a strong preference for the GC and PLV connectivity measures, closely followed by absCPCC. This preference for GC is expected because it provides a comprehensive description of brain activity without separating information by frequency band. Additionally, GC is directed, meaning that one electrode influencing another does not imply the reverse, and it generates twice the number of features as undirected methods. As for PLV, it has been previously demonstrated to effectively capture changes due to rehabilitation [20,21], which is consistent with our results.

Unexpected findings emerged when considering connectivity measures that were designed to avoid the effects of volume conduction. These measures are wPLI and iCPCC. The characteristic property of volume conduction is that signals detected at two or more electrodes are in phase. In contrast to this, the signals of neural communication are transferred via axons with a limited speed of up to 120 m/s, which produces a propagation delay and a phase lag between signals detected at two electrodes. Connectivity measures can avoid volume conduction by rejecting signal components that are in phase. This approach, however, degrades short connections with a small phase lag. The effect is more pronounced for lower frequencies, which complies with our results, where the accuracy obtained using wPLI and iCPCC gradually increases for higher frequency bands. Overall, our results show the high importance of short-range connections in modeling the impact of rehabilitation on stroke patients, since nearby areas often take over the function of damaged regions [12,20].

The results obtained for the MI connectivity measure are also highly interesting. MI performs best among all band-limited methods in the theta band, with median accuracy of 90.8% and average accuracy of 85.8%, while, overall, its performance varies considerably. The theta band performance of MI is comparable to the best PLV and GC results. However, in the theta band, PLV and other phase-lag-based methods perform worse than in other frequency bands. The theta band activity is associated with memory tasks and with speech production [39]. As such, it seems crucial in conveying information about neural adaptation to stroke. However, the lower performance of phase-lag-based methods and the better performance of MI, which is a measure of information transfer, may imply that phase locking is not a critical characteristic of brain reorganization after a stroke. Conversely, the relative success of MI features in this band suggests that the adaptation to damage and rehabilitation after a stroke leads to significant changes in the (inter)dependency, information flow, and organization of neural oscillations in the theta band, independent of phase locking.

Another surprising set of findings was the lack of a difference in accuracy between data from verbal and spatial working memory tasks. We initially anticipated better results with data from verbal tasks, considering that Broca’s aphasia primarily involves difficulties in speech production [5]. However, our classifier performed similarly for both tasks, suggesting that post-rehabilitation brain reorganization may not strongly depend on the site of damage or be closely tied to specific functions. Instead, it seems that improvements in overall brain function, such as global connectivity, restructuring, and the establishment of alternative routes for information flow around damaged areas, lead to enhancements in all functions, irrespective of the primary symptoms after a stroke.

Our findings unexpectedly revealed no significant difference in classifier accuracy between verbal and spatial working memory tasks. This was particularly surprising given our initial hypothesis of superior performance with verbal task data, considering the speech production challenges inherent in Broca’s aphasia [5]. The similar classifier performance across both task types could suggest that the post-rehabilitation brain reorganization might extend beyond the directly affected areas, potentially involving more global processes of connectivity and restructuring. These results hint at a more complex interplay of rehabilitative improvements, where the enhancement of overall brain function, including the establishment of compensatory information pathways, may contribute to recovery across various functions, not just those primarily impacted by stroke.

An alternative interpretation might provide additional insight. Broca’s area, while primarily known for its role in speech production, is also crucial for working memory across domains (verbal and non-verbal) [21]. It is conceivable that rehabilitation, although focused on speech, can compensate for the damage to this area by enhancing the corresponding contralateral regions in the brain. This compensation could thus improve working memory in general. Such a perspective could suggest that improvements in speech post-rehabilitation may be partly attributed to enhanced working memory capabilities, facilitated by the brain’s adaptability and the establishment of new neural pathways. Further research is needed to explore this finding and its implications for stroke rehabilitation, particularly in the context of specific language impairments like Broca’s aphasia.

### Limitations

The primary limitation of our study is the small size of the dataset. Despite our efforts to ensure subject-invariant classification, the limited number of subjects may introduce unexpected biases. Additionally, the small sample size restricts our ability to detect smaller yet potentially significant effects, which might go unnoticed in statistical analysis. The feature selection process also does not account for statistical dependencies between features, which could result in redundant feature use and increase the complexity of the model. In future works, the use of minimum redundancy maximum relevance can be beneficial to avoid this issue. Despite these limitations, we believe that our study has important implications for both the machine learning and neuroscience communities. It demonstrates that even with limited data, meaningful results can be obtained, illustrating how different methods of characterizing brain connectivity affect the accuracy of tracking the process of rehabilitation in Broca’s aphasia.

## 5. Conclusions

To conclude, our study succeeded in creating an accurate classifier of pre- and post-rehabilitation states, based on connectivity measures of EEG data during verbal and spatial working memory tasks. We found that the choice of connectivity measure to be used as an input feature greatly affected the accuracy of our classifier, with the methods designed to eliminate volume conduction performing the worst. Surprisingly, the classifier performed equally well with the data recorded during a verbal or spatial working memory task. Furthermore, the results of classification suggest that different connectivity measures represent distinct neural processes taking place during rehabilitation for Broca’s aphasia. While further work has to be done to increase the explainability and validity of machine learning models, we hope that this work will serve as a stepping stone to the systematical evaluation of neural correlates of rehabilitation, using machine learning as a tool of discovery. This paper distinguishes itself from other available studies by employing a novel combination of advanced EEG analysis, functional connectivity metrics, and machine learning to specifically address the rehabilitation of Broca’s aphasia following a stroke. This focused strategy, paired with a meticulous methodological design that enables meaningful results even with limited data, sets it apart in the realm of stroke rehabilitation research.

## Figures and Tables

**Figure 1 sensors-24-00329-f001:**
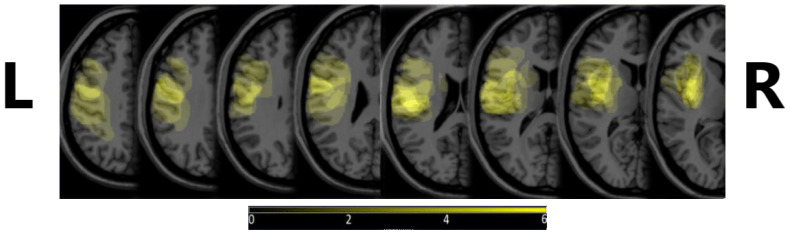
Lesion density maps for our subjects. A more intense yellow color denotes the more frequent occurrence of a lesion in that area. The brightest yellow shows that 6 or more subjects had a lesion there. L and R letters denote the left and right side of the head, respectively, to avoid confusion with MRI image display conventions.

**Figure 2 sensors-24-00329-f002:**
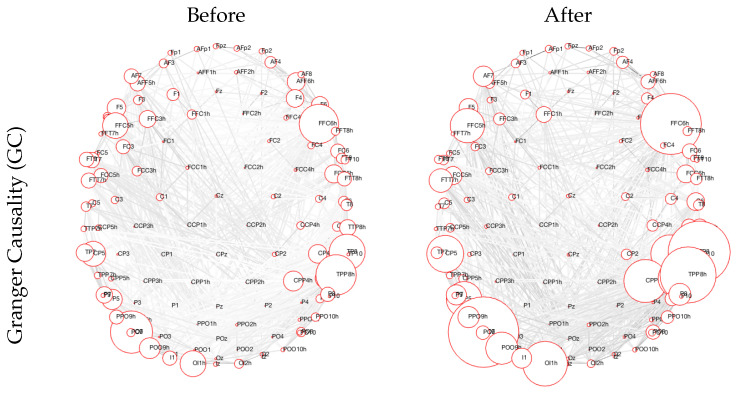
The 10% most significant changes in Granger causality (GC) before and after rehabilitation for Broca’s aphasia, during a verbal working memory task. The size of the node indicates a stronger correlation with other nodes, whereas the color of the lines connecting the electrodes can be ignored—the color hues are used for visualization purposes only.

**Figure 3 sensors-24-00329-f003:**
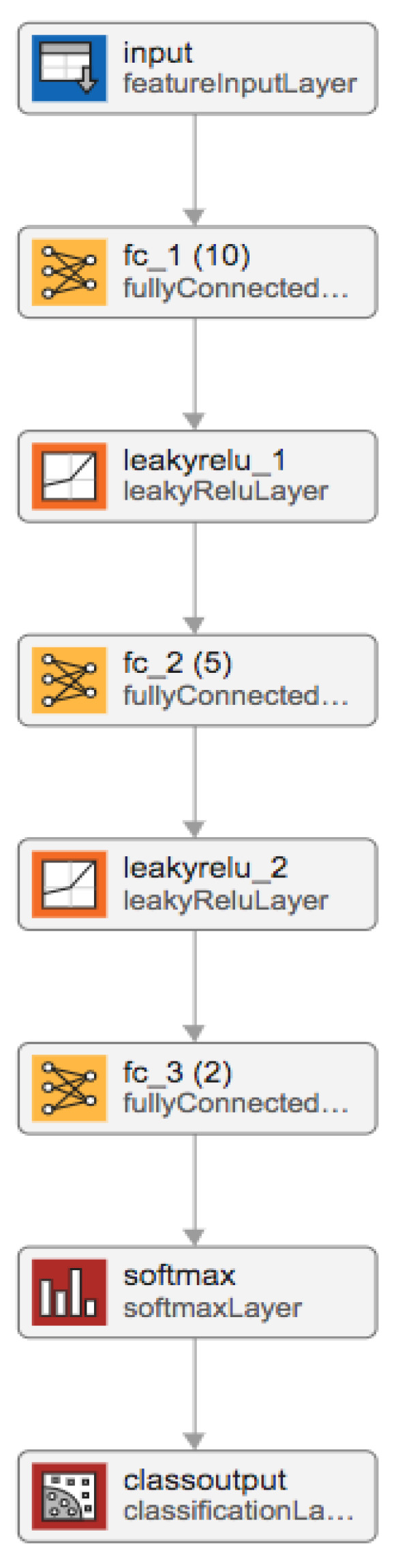
Machine learning model layers.

**Figure 4 sensors-24-00329-f004:**
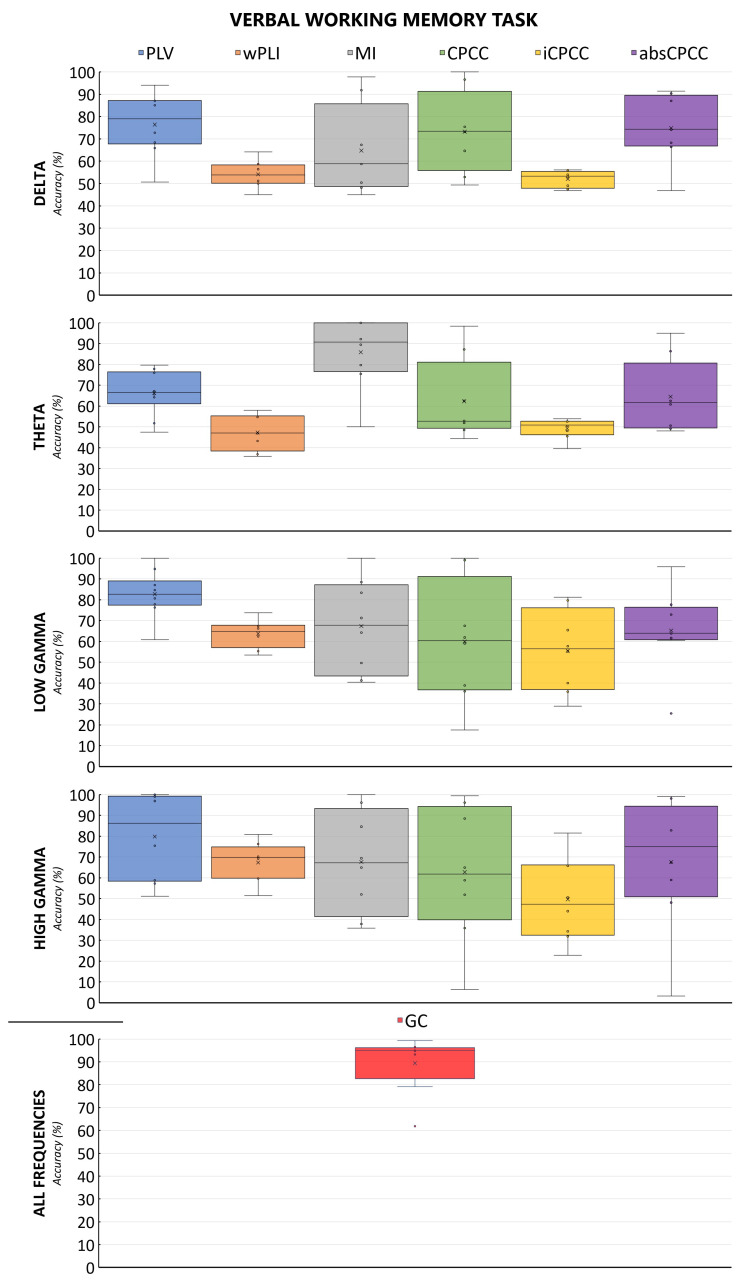
Accuracies and their distributions for classification of EEG to distinguish the state before and after the rehabilitation for the verbal memory task. Each column in the same color represents a connectivity measure, computed for frequency bands shown in rows. Dots on the graph represent individual data points, i.e., accuracies obtained in the leave-one-out cases.

**Figure 5 sensors-24-00329-f005:**
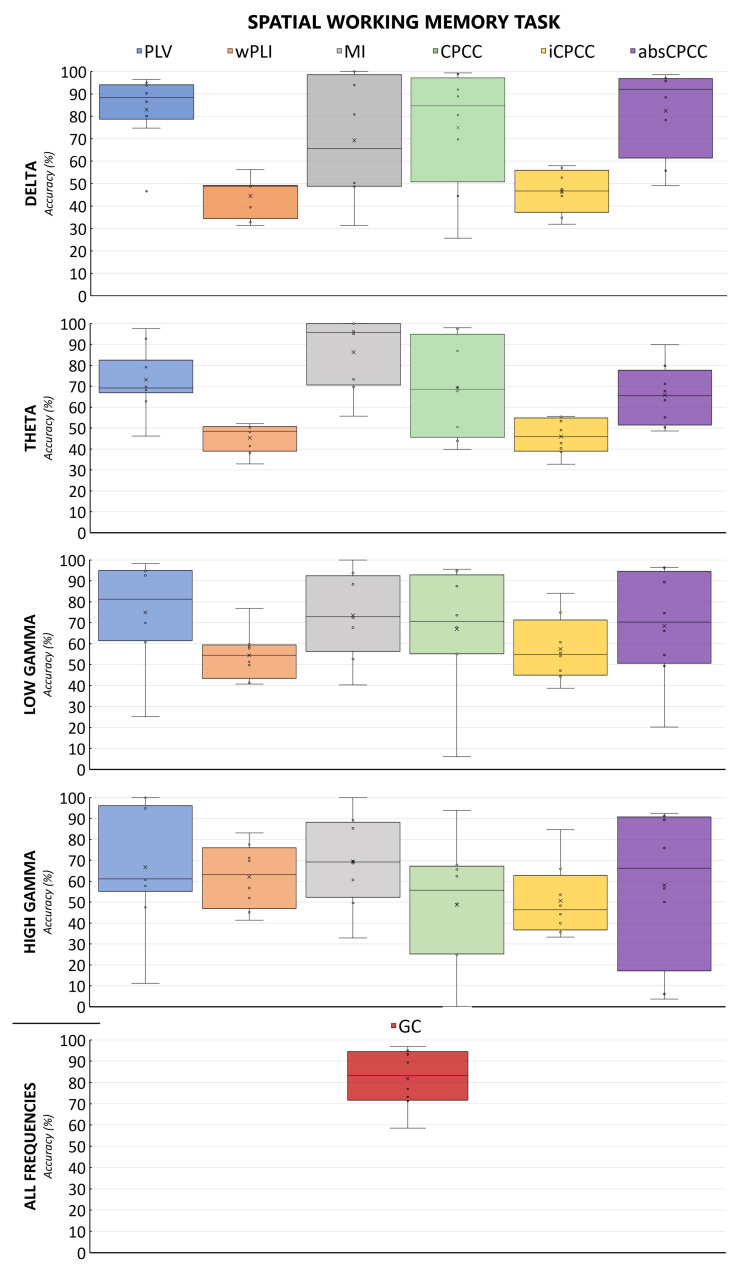
Accuracies and their distributions for classification of EEG to distinguish the state before and after the rehabilitation for the spatial memory task. Each column in the same color represents a connectivity measure, computed for frequency bands shown in rows. Dots on the graph represent individual data points, i.e., accuracies obtained in the leave-one-out cases.

**Figure 6 sensors-24-00329-f006:**
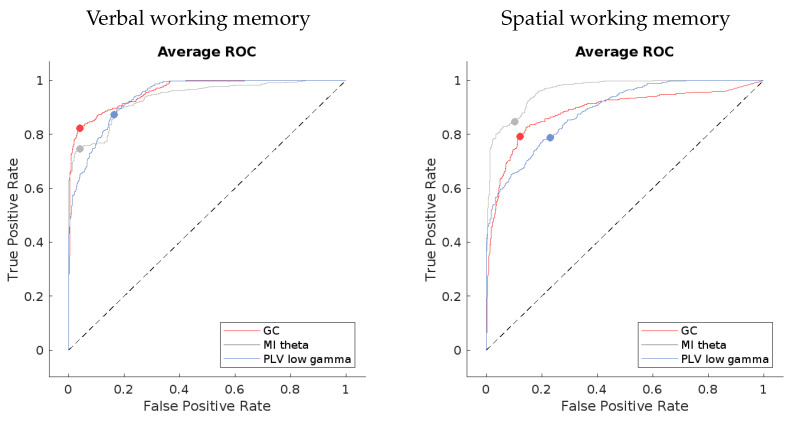
Average ROC curves for the best-performing methods when using verbal working memory data on the left and spatial working memory data on the right. Every line of a certain color represents a method–frequency combination being classified in the post-rehabilitation class.

**Figure 7 sensors-24-00329-f007:**
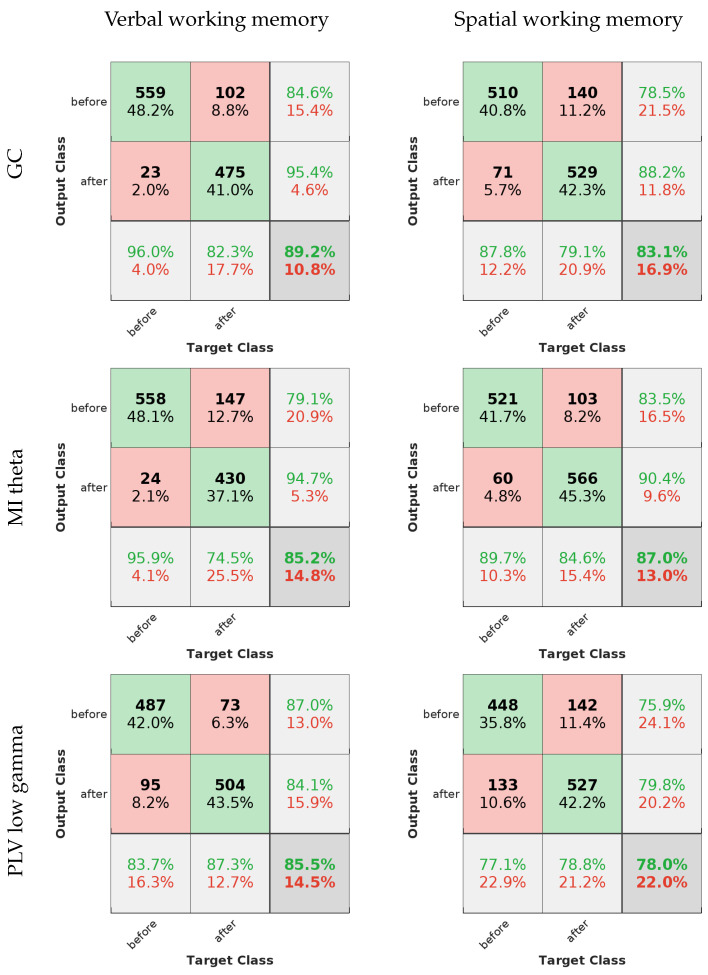
Mean confusion matrices for the best-performing methods, using verbal working memory data on the left and spatial working memory data on the right.

**Table 1 sensors-24-00329-t001:** Descriptive data on accuracies achieved by our machine learning model. All the accuracies are shown in percentages [%]. Shading with darker gray highlights the best method–frequency combination for each of the connectivity measures. Please note that the best here refers to the ones that allowed for the most accurate classification of verbal working memory data, as this was the main focus of the present study.

	Verbal Working Memory Task			Spatial Working Memory Task		
	**Mean**	**Med.**	**Std.**	**Mean**	**Med.**	**Std.**
GC all	89.4	95.0	12.7	81.8	83.2	13.8
PLV delta	76.4	78.9	14.5	82.9	88.4	16.5
PLV theta	66.2	66.4	11.8	73.1	69.2	16.5
PLV low gamma	82.7	82.7	12.1	74.8	81.3	25.5
PLV high gamma	79.8	86.2	21.6	66.7	61.2	30.7
wPLI delta	54.1	53.8	6.0	44.5	48.9	8.9
wPLI theta	47.3	47.0	8.4	45.3	48.4	7.0
wPLI low gamma	63.7	64.7	6.7	54.5	54.5	11.6
wPLI high gamma	67.3	69.7	9.5	62.1	63.2	15.4
MI delta	64.8	58.9	19.9	69.3	65.6	27.4
MI theta	85.8	90.8	17.3	86.2	95.6	17.4
MI low gamma	67.3	67.7	22.4	73.5	72.8	22.4
MI high gamma	67.6	67.1	24.7	69.5	69.1	22.0
CPCC delta	73.2	73.4	18.2	74.9	84.7	26.9
CPCC theta	62.3	52.8	19.7	69.3	68.7	23.3
CPCC low gamma	60.0	60.4	29.3	66.9	70.6	29.4
CPCC high gamma	62.7	61.9	32.0	48.7	55.7	30.1
iCPCC delta	52.0	53.3	3.2	46.5	46.7	9.5
iCPCC theta	49.3	50.8	5.3	46.0	45.9	9.2
iCPCC low gamma	55.5	56.4	29.2	57.4	54.8	16.6
iCPCC high gamma	49.7	47.3	26.8	50.7	46.3	17.3
absCPCC delta	74.9	74.4	15.7	82.4	92.1	20.3
absCPCC theta	64.5	61.7	18.8	65.7	65.5	14.4
absCPCC low gamma	65.2	63.9	26.0	68.4	70.3	25.9
absCPCC high gamma	67.6	75.1	19.4	58.1	66.2	41.9

**Table 2 sensors-24-00329-t002:** Machine learning evaluation methods. Precision, sensitivity, specificity, accuracy, and F-measure for the post-rehabilitation class for the verbal working memory data on the left and the spatial working memory data on the right. Precision is the number of correctly classified true positives out of all positives. Sensitivity is the ratio between the correctly classified positives and the total number of true positives and false negatives. Specificity is the ratio between the true negatives and the total number of false positives and true negatives. The F-measure or F1 score is used similarly to accuracy but tests different aspects of a machine learning model—false negatives and false positives.

	Verbal Working Memory			Spatial Working Memory		
	**GC**	**MI Theta**	**PLV Low Gamma**	**GC**	**MI Theta**	**PLV Low Gamma**
Precision	0.95	0.95	0.84	0.88	0.90	0.80
Sensitivity	0.82	0.75	0.87	0.79	0.85	0.79
Specificity	0.96	0.96	0.84	0.88	0.90	0.77
Accuracy	0.89	0.85	0.86	0.83	0.87	0.78
F-measure	0.88	0.83	0.86	0.83	0.87	0.79

**Table 3 sensors-24-00329-t003:** Results of the omnibus ordinary two-way ANOVA, examining whether any differences in accuracy of the classifier occurred due to the data coming from different tasks being performed (row 1), the connectivity measure chosen as the input (row 2), or an interaction between the two (row 3). Intercept and error rows represent general information about the ANOVA model used.

Source	Type III Sum of Squares	df	Mean Square	F	Sig.	Partial Eta Squared
Task	145.194	1	145.194	0.085	0.775	0.006
Connectivity measure	51,912.160	24	2163.007	6.384	0.000	0.313
Interaction	4486.344	24	186.931	0.552	0.959	0.038
Intercept	1,709,417.789	1	1,709,417.789	1000.735	0.000	0.986
Error (task)	23,914.269	14	1708.162			
Error (connectivity measure)	113,834.074	336	338.792			

**Table 4 sensors-24-00329-t004:** Results of the post-hoc parametric testing of classification accuracy, comparing connectivity measures used as input features. All column values, except for the *p*-value, are in % of accuracy. Diff. stands for the difference between the accuracy of the model with the connectivity measure in the leftmost column and the model with the connectivity measure in the next (second column). CI is the confidence interval of the difference (Bonferroni-corrected 95% interval). All *p*-values are corrected for the total number of pairwise comparisons possible.

		Diff.	Std. Error	*p*-Value	CI Lower Bound	CI Upper Bound
GC	PLV theta	15.956	2.993	0.032	0.739	31.173
	wPLI delta	36.309	3.006	0.000	21.027	51.592
	wPLI theta	39.329	3.589	0.000	21.082	57.575
	wPLI low gamma	26.551	4.517	0.012	3.590	49.512
	iCPCC delta	36.351	3.328	0.000	19.434	53.267
	iCPCC theta	38.014	4.026	0.000	17.549	58.478
	iCPCC low gamma	29.177	4.713	0.007	5.217	53.136
	iCPCC high gamma	35.447	4.522	0.001	12.460	58.435
PLV delta	wPLI delta	30.349	2.992	0.000	15.138	45.560
	wPLI theta	33.369	3.652	0.000	14.803	51.934
	iCPCC delta	30.390	3.352	0.000	13.352	47.428
	iCPCC theta	32.053	4.154	0.001	10.935	53.171
	iCPCC low gamma	23.217	4.223	0.024	1.749	44.684
	iCPCC high gamma	29.487	4.709	0.006	5.548	53.426
PLV theta	wPLI delta	20.354	3.207	0.005	4.049	36.659
	wPLI theta	23.373	3.050	0.001	7.867	38.879
	iCPCC delta	20.395	3.208	0.005	4.088	36.702
	iCPCC theta	22.058	3.856	0.016	2.457	41.659
	iCPCC high gamma	19.492	3.194	0.008	3.256	35.728
PLV low gamma	wPLI delta	29.468	5.088	0.014	3.605	55.330
	wPLI theta	32.487	5.189	0.006	6.108	58.866
	iCPCC delta	29.509	5.583	0.035	1.126	57.891
	iCPCC theta	31.172	5.967	0.039	0.837	61.507
wPLI delta	MI theta	−36.725	4.529	0.000	−59.747	−13.703
	absCPCC delta	−29.336	3.616	0.000	−47.719	−10.952
wPLI theta	wPLI high gamma	−18.405	3.376	0.026	−35.569	−1.242
	MI theta	−39.745	4.798	0.000	−64.137	−15.352
	CPCC delta	−27.761	5.454	0.049	−55.489	−0.033
	absCPCC delta	−32.355	4.268	0.001	−54.052	−10.659
wPLI low gamma	iCPCC theta	11.463	1.968	0.013	1.456	21.469
wPLI high gamma	iCPCC delta	15.427	2.719	0.017	1.607	29.248
	iCPCC theta	17.090	2.566	0.003	4.047	30.134
MI theta	iCPCC delta	36.766	4.311	0.000	14.851	58.681
	iCPCC theta	38.429	4.219	0.000	16.983	59.875
	iCPCC low gamma	29.593	4.938	0.010	4.491	54.694
	iCPCC high gamma	35.863	5.722	0.006	6.777	64.949
CPCC delta	iCPCC delta	24.782	4.478	0.022	2.019	47.546
iCPCC delta	absCPCC delta	−29.377	3.758	0.001	−48.483	−10.271
absCPCC delta	iCPCC theta	31.040	3.937	0.000	11.026	51.054
	iCPCC high gamma	28.474	5.382	0.034	1.113	55.835
iCPCC high gamma	absCPCC delta	−28.474	5.382	0.034	−55.835	−1.113

## Data Availability

The raw data supporting the conclusions of this article will be made available by the authors on request, for the purposes of replication and validation of the results.

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
