# Peer review of "Integrating EEG and Machine Learning to Analyze Brain Changes during the Rehabilitation of Broca’s Aphasia"

_sensors, 2024, doi:10.3390/s24020329_

Round 1

Reviewer 1 Report

Comments and Suggestions for Authors

The paper aimed to design a precise machine learning model to analyze changes in brain in the process of rehabilitation after stroke for eight patients of Brocas aphasia. Functional connectivity analysis was performed employing

various metrics such as Granger Causality (GC), Phase-Locking Value (PLV), weighted Phase-Lag Index (wPLI), Mutual Information (MI), and Complex Pearson Correlation Coefficient (CPCC). The connectivity estimates were used as an input to a neural network classifier, trained to distinguish between pre- and post- rehabilitation states. Generally, the topic is very interesting. The novelty and contribution of the paper is vague. Also, several comments need to be addressed.

in the abstract. Could you please highlight the novelty and main contributions? Also please add the numerical findings of the study

The introduction section is too long and contains many background information.

There is no related work section which is very important in any scientific article. Please add related work that employed machine learning to analyze changes in the brain in the process of rehabilitation after a stroke.

Please highlight the novelty and contribution.

Please add a paragraph stating the organization of the paper by the end of the introduction.

Why did the author use only EEG of only 8 patients?

What are the exclusion criteria for the experiment?

How many subjects participated in face cues data acquisition?

More details regarding the dataset acquisition should be added.

What are the class labels for each of the data types?

Please mention the size of each feature set you extracted.

What are the size of the feature matrix?

Please mention the type of the deep learning model that the author used. The authors should have used LSTM or Bi-LSTM, or recurrent networks

What are the class labels of the EEG data?

Please define the performance evaluation methods.

Please add the precision, sensitivity, specificity, and F1 measure as they are important metrics for evaluating models

Please add roc curves and confusion matrices.

Please add your limitations

Comments on the Quality of English Language

There are some punctuation errors

Author Response

Response to Reviewer 1’s Comments

Yes

Can be improved

Must be improved

Not applicable

Does the introduction provide sufficient background and include all relevant references?

( )

( )

(x)

( )

Are all the cited references relevant to the research?

( )

( )

(x)

( )

Is the research design appropriate?

( )

( )

(x)

( )

Are the methods adequately described?

( )

( )

(x)

( )

Are the results clearly presented?

( )

( )

(x)

( )

Are the conclusions supported by the results?

( )

( )

(x)

( )

The paper aimed to design a precise machine learning model to analyze changes in brain in the process of rehabilitation after stroke for eight patients of Broca’s aphasia. Functional connectivity analysis was performed employing various metrics such as Granger Causality (GC), Phase-Locking Value (PLV), weighted Phase-Lag Index (wPLI), Mutual Information (MI), and Complex Pearson Correlation Coefficient (CPCC). The connectivity estimates were used as an input to a neural network classifier, trained to distinguish between pre- and post- rehabilitation states. Generally, the topic is very interesting. The novelty and contribution of the paper is vague. Also, several comments need to be addressed.

We thank the reviewer for their detailed and in-depth comments on our work. We have edited our manuscript according to Your comments to the best of our ability, with all alterations being marked in orange color and listed in the point-by-point response with their corresponding line numbers. Likewise, we believe that the manuscript is much improved by taking these into account and are happy to be given the chance to do so.

in the abstract. Could you please highlight the novelty and main contributions? Also, please add the numerical findings of the study

We have rewritten the abstract (lines 1-18), to better reflect the novelty and main contributions of our paper. We have added the three most important numerical results as well. 

The introduction section is too long and contains many background information.

The introduction section was appropriately shortened, and only limited background information was retained (lines 20-57, 60-62, 69-74, 85-108 and 120-141).

There is no related work section which is very important in any scientific article. Please add related work that employed machine learning to analyze changes in the brain in the process of rehabilitation after a stroke.

A related works section was added to the introduction (lines 90-106), with two studies that used EEG-derived measures to classify either the rehabilitation status or success presented therein. To the best of our knowledge, these are the only two studies that use EEG data and attempt to predict recovery from aphasia (either its success or presence), using machine learning (as opposed to statistics in the classical sense).

Please highlight the novelty and contribution.

The novelty and contribution of our paper is highlighted in the “Abstract” (lines 11-18), “Related works” section of the introduction (lines 103-106), discussion (lines 480-485 and 558-561) and conclusion (lines 573-578).

Please add a paragraph stating the organization of the paper by the end of the introduction.

A section detailing the organization of our paper was added to the tail end of the introduction (lines 107-141).

Why did the author use only EEG of only 8 patients?

We wholeheartedly agree with the reviewer that the sample size of 8 patients is small. However, limited data is common in this type of research. While the sample size is a limitation, the rigorous leave-one-out cross-validation approach maximizes the use of available data and enhances the generalizability of the model. This explanation was also added to the “Materials and Methods” section (lines 179-183)

 What are the exclusion criteria for the experiment?

 We added a section with more detailed inclusion/exclusion criteria, as per your request (lines 161-170).

How many subjects participated in face cues data acquisition?

We kindly ask for further clarification on this comment. In the present study, no face cues were presented, neither are they mentioned in our paper. We have used the data of eight patients, as stated in the first paragraph of the methods section (line 146). We hope the methods section adequately states the number of participating subjects, but we are happy to make any changes required after receiving further explanation.

More details regarding the dataset acquisition should be added.

More details were added, including lesion site data regarding the patients (lines 161-170, Figure 1).  

What are the class labels for each of the data types?

We have added an explicit statement of what the class labels are under the »Statistical feature selection« section (lines 309-322) and added the explanation to the section “Functional connectivity calculation” (line 288), where mention of the class first appears in the text. We have also improved the clarity of the sentence explaining the class targets of our machine learning model under the section “Our machine learning model” (line 338).

Please mention the size of each feature set you extracted.

We added listings of the number of values in each connectivity matrix, that were used to statistically extract the most prominent features, under the section “Functional connectivity calculation” (lines 297-300).

What are the size of the feature matrix?

We added specific numbers of elements of each matrix, where the final features were extracted from (lines 298-300). We have also added the specific number of features under the section “Statistical feature selection” (line 314). 

Please mention the type of the deep learning model that the author used. The authors should have used LSTM or Bi-LSTM, or recurrent networks.

We thank the reviewer for their constructive comment, but our reasoning for the approach used was thus; connectivity features are neuroscientificially meaningful indicators for assessing the rehabilitation after stroke. Recognizing their significance, we have chosen to use and analyze these features in our study. However, it's essential to note that connectivity features encapsulate an entire epoch, lacking insights into temporal dynamics. On the other hand, being relatively compact, we can classify them using a neural network with only three fully connected layers, as shown in Figure 3 (previously Figure 2). While Long Short-Term Memory (LSTM) or Bidirectional LSTM (Bi-LSTM) networks excel in analyzing signals over time, they may not be optimal for classifying connectivity features. An alternative approach involves utilizing these networks directly on raw temporal EEG data. Nonetheless, this strategy may necessitate a substantially larger training dataset than used in the current study. We hope this answers Your consideration adequately.

Please add the precision, sensitivity, specificity, and F1 measure as they are important metrics for evaluating models

We have added these characteristics of our models for the three top performing connectivity measure inputs used to the Results section, subsection “Evaluation of the best-performing methods” (lines 385-415). 

Please add roc curves and confusion matrices.

We have added the ROC curves and confusion matrices for the three top performing connectivity measure inputs used to the Results section, subsection “Evaluation of the best-performing methods” (lines 385-415).

Please add your limitations

We have reorganized our discussion to more clearly present our limitations in the last paragraph. For clarity, we have also added a subsection keyword for this at the start, to make it more easily accessible. We have added another explanation of a limitation, as expressed by another reviewer, to the existing paragraph on the limitations of our study (lines 549-561).

There are some punctuation errors

English language has been thoroughly edited throughout the manuscript.

Reviewer 2 Report

Comments and Suggestions for Authors

See comments in the attached document.

Author Response

Response to Reviewer 2’s Comments

Yes

Can be improved

Must be improved

Not applicable

Does the introduction provide sufficient background and include all relevant references?

(x)

( )

( )

( )

Are all the cited references relevant to the research?

(x)

( )

( )

( )

Is the research design appropriate?

( )

( )

(x)

( )

Are the methods adequately described?

( )

(x)

( )

( )

Are the results clearly presented?

( )

( )

(x)

( )

Are the conclusions supported by the results?

( )

( )

(x)

( )

The aim of this paper is to analyze changes in brain in the process of rehabilitation after stroke by means of a machine learning model applied of EEG signals. For that, authors use a neural network as a classifier.

We sincerely thank the reviewer for their helpful comments to our manuscript. We believe our paper is much improved by these corrections. All the changes to the text (in response to comments from all three reviewers) are marked in orange, and listed with their corresponding line numbers in the point-by-point response.

However, instead of using raw EEG signals, they choose several metrics as input. A first drawback is probably the use of a Neural Network as a classifier with that kind of inputs, but it is feasible.

We thank the reviewer for their constructive criticism, but we have decided to use the approach detailed in the paper as connectivity features are neuroscientificially meaningful indicators for assessing the rehabilitation after stroke. Recognizing their significance, we have chosen to use and analyze these features in our study. Due to being relatively compact, we can classify them using a neural network with only three fully connected layers, as shown in Figure 3 (previously Figure 2). The alternative approach of using neural networks on raw EEG data would require a more complex architecture and, as such, also substantially larger training dataset. We have added some additional explanation as to our rationale for this choice to the section “Functional connectivity calculation” (lines 301-307). We hope this answers Your consideration adequately.

The main drawback is the statistical feature selection done in which authors compare two by two the metrics previously chosen and they not take into account the dependencies among features. A method like minimum redundance maximum relevance (mRMR) would be more reliable since it analyses the dependency between the joint distribution of the selected features and the classification variable.

We wholeheartedly agree with the reviewer that the feature selection method is simplistic and has its drawbacks. To reflect this, we have added the statement of this to our “Limitations” section (lines 553-557). However, We selected features using simple statistics, as this offers the intuitive explanation that only the features that change the most are kept, which, regardless of their interdependence, enables easier discovery of brain activity characteristics in the future.

Indeed, training and validation of the model was done separately for each combination of features, what I understand as an avoidance of analysis of dependencies among features before applying the classification model.

We thank the reviewer for recognizing the merits of our approach. Additionally, the measures were treated separately to enable comparisons of which measure is the most suited for use in our model, based on the accuracy achieved. 

Regarding the experimental analysis, 8 patients is not too many, but it is usual in this kind of research, and a leave-one-out by patient is a good approach to analyze the capabilities of generalization of the model.

We again thank the reviewer for their recognition. We have added a sentence explaining this explicitly in the section “EEG data acquisition” (lines 179-183).

A subsection of experimental analysis explaining all the experiments with the corresponding metrics and another one with the results would be clearer.

We have restructured our results and added more data on model performance, which is now contained in its own section of the results “Evaluation of the best-performing methods” (lines 385-415). We hope this sufficiently improved the clarity of our results.  

As well, instead of showing min and max, it would be better to show the ranges (mean +- std) and also the accuracies by each patient

Table 1 has been edited, and now displays the mean and SD, instead of min-max range. We added a detailed breakdown of the three best models’ performance metrics, including ROC curves, confusion matrices and precision, sensitivity, specificity, and F-measure tables to the paper. We hope this sufficiently expands the data on the accuracy of our model.

Reviewer 3 Report

Comments and Suggestions for Authors

This study explores the integration of electroencephalography (EEG) and machine learning for rehabilitation research, specifically focusing on eight patients with Broca's aphasia who underwent stroke rehabilitation. The subjects performed verbal and visuospatial working memory tasks before and after the treatment. The researchers designed a machine learning model using the following connectivity metrics: Granger Causality (GC), Phase-Locking Value (PLV), weighted Phase-Lag Index (wPLI), Mutual Information (MI), and Complex Pearson Correlation Coefficient (CPCC). The neural network classifier, trained on these metrics, demonstrated high accuracy levels exceeding 90% in distinguishing between pre- and post-rehabilitation states for GC and PLV-based features. Moreover, the classifier provided similar results for both tasks.

The topic is interesting but the paper needs to be improved. I have the following comments and suggestions:

1)    In the introduction most of the references are related to aphasia. As you have described the recovery process of the brain after stroke, add more references related to the recovery process in general.

2)    Define all abbreviations in the main text before using them.

3)   Cite the references [14,17] only once to improve the readability (lines 102,104).

4)    Add references about neurofeedback (lines 128-130).

5)    Add more details about the patients (demographic data and the site of the stroke lesion).

6)    Add more information about PLV and wPLI in the methods section.

7)  Explain why the delta, theta and gamma bands (not alpha and beta) were chosen for the analysis (lines 261-262).

8) If the approach used for the feature selection has been already employed in literature, please cite the references (Section 2.4).

9)    Add the version of Matlab used.

10) In order to improve the readability of Table 1, highlight for each group of methods the band at which the best performance is obtained.

11) Figure 1 should be modified to be clearer and more informative.

12) Divide the description of the results into paragraphs, one for each method.

13) What is the novelty of your study compared to the previous ones? Stress it in the paper and add a comparison with them.

Comments on the Quality of English Language

Minor editing of English language is required.

Author Response

Response to Reviewer 3’s Comments

Yes

Can be improved

Must be improved

Not applicable

Does the introduction provide sufficient background and include all relevant references?

( )

(x)

( )

( )

Are all the cited references relevant to the research?

( )

( )

(x)

( )

Is the research design appropriate?

( )

(x)

( )

( )

Are the methods adequately described?

( )

( )

(x)

( )

Are the results clearly presented?

( )

( )

(x)

( )

Are the conclusions supported by the results?

( )

(x)

( )

( )

This study explores the integration of electroencephalography (EEG) and machine learning for rehabilitation research, specifically focusing on eight patients with Broca's aphasia who underwent stroke rehabilitation. The subjects performed verbal and visuospatial working memory tasks before and after the treatment. The researchers designed a machine learning model using the following connectivity metrics: Granger Causality (GC), Phase-Locking Value (PLV), weighted Phase-Lag Index (wPLI), Mutual Information (MI), and Complex Pearson Correlation Coefficient (CPCC). The neural network classifier, trained on these metrics, demonstrated high accuracy levels exceeding 90% in distinguishing between pre- and post-rehabilitation states for GC and PLV-based features. Moreover, the classifier provided similar results for both tasks.

The topic is interesting but the paper needs to be improved. 

We sincerely thank the reviewer for their constructive comments on our manuscript and the interest in our work. We believe the modifications to the paper in line with Your comments improved it significantly. All changes to the document are marked in orange, with corresponding line numbers listed in this point-by-point response.

I have the following comments and suggestions:

1)    In the introduction most of the references are related to aphasia. As you have described the recovery process of the brain after stroke, add more references related to the recovery process in general.

We have added more references describing the recovery process in general. We added 3 new references, Kiran 2019, Wilson 2022 and Stockert 2020. In the “References”, these are the 6., 7. and 15. reference.

2)    Define all abbreviations in the main text before using them.

We parsed the text and corrected this issue wherever we found it.

3)   Cite the references [14,17] only once to improve the readability (lines 102,104).

This part of the text has been rewritten in line with other reviewer’s comments. The consideration for readability was taken into account, and the same sources are not cited multiple times in a single paragraph.

4)    Add references about neurofeedback (lines 128-130).

This part of the text has been removed to shorten the introduction and improve the overall readability of the paper, as suggested by other reviewers. 

5)    Add more details about the patients (demographic data and the site of the stroke lesion).

We added more data about the patients, including the site of the stroke lesion (Figure 1 and lines 161-170).

6)    Add more information about PLV and wPLI in the methods section.

We expanded the descriptions of the connectivity measures we used, with the explanation of what characteristic of the EEG signal they pertain to (lines 248-253, 259-260, 264-267).

7)  Explain why the delta, theta and gamma bands (not alpha and beta) were chosen for the analysis (lines 261-262).

We added an explanation for the selection of these bands. In short, alpha and beta are more prone to being influenced by the fact that both of our tasks involve working memory and might thus be confounded by interpersonal variability in the ability to perform this task, regardless of the baseline condition (would not reflect the changes due to rehabilitation as clearly). Moreover, these bands are less consistently associated with language. Thus, we decided to use the measures that will give us the greatest chance of providing a clear, language-related input to the classifier (lines 279-285).

8) If the approach used for the feature selection has been already employed in literature, please cite the references (Section 2.4).

We added a reference to a paper that uses a similar approach for feature selection (reference 31).

9)    Add the version of Matlab used.

The version of MATLAB we used was added (line 329).

10) In order to improve the readability of Table 1, highlight for each group of methods the band at which the best performance is obtained.

We highlighted the best performing band for each method, as requested. An explanation of this was also added to the table caption. 

11) Figure 1 should be modified to be clearer and more informative.

We have removed the PLV data and now only showcase the GC data, to provide an example of connectivity change. Clarity should thus be improved, as well as simplicity of the figure.

12) Divide the description of the results into paragraphs, one for each method.

As we have used many methods of connectivity estimation, in combination with four frequency bands, we instead present model performance data for the three best performing methods in its own section of the results “Evaluation of the best-performing methods” (lines 385-415). As commentary on each method-frequency band pair would most likely significantly lengthen the main body of the text, we omit other data on this topic. We hope this solution is satisfactory and sufficiently improves the clarity of our work.

13) What is the novelty of your study compared to the previous ones? Stress it in the paper and add a comparison with them.

We have added statements of novelty to the abstract, introduction, discussion and conclusion sections (lines 11-18, 103-106, 480-485 and 558-561 and 573-578). We also discuss differences with previous related works in the discussion (lines 473-485), pointing out the novelty of our approach to this problem.

 Minor editing of English language is required.

English language has been thoroughly edited throughout the text.

Round 2

Reviewer 1 Report

Comments and Suggestions for Authors

Thank you for addressing my comments. However, the related work section should include more papers

Author Response

Response to Reviewer 1’s comments

Yes

Can be improved

Must be improved

Not applicable

Does the introduction provide sufficient background and include all relevant references?

(x)

( )

( )

( )

Are all the cited references relevant to the research?

( )

(x)

( )

( )

Is the research design appropriate?

(x)

( )

( )

( )

Are the methods adequately described?

(x)

( )

( )

( )

Are the results clearly presented?

(x)

( )

( )

( )

Are the conclusions supported by the results?

(x)

( )

( )

( )

Comments and Suggestions for Authors:

We thank the reviewer for their kind assistance with improving our manuscript. The changes made can be seen marked in red, with the point-by-point responses provided below.

Thank you for addressing my comments. However, the related work section should include more papers.

We thank the reviewer for encouraging us to include more related works in our manuscript. We found two review articles, one from August 2023, which review the use of EEG in predicting rehabilitation status and/or success. From the therein gathered studies and our own search for related works using Google Scholar (search string; connectivity, rehabilitation, EEG, “machine learning”, aphasia, language, -motor, -schizophrenia), we find three more studies, which are somewhat adjacent to our work and whose results we can compare to those we obtained. The criteria for the inclusion of the studies as work related to our was that the study focused on EEG data, attempted to either classify or discriminate between before and after rehabilitation state for aphasia as a result of stroke using this data, or to determine the success of such rehabilitation. We prioritized the studies using modern machine learning approaches and connectivity measures. We added the descriptions of these studies to the introduction (lines 102-130) and compared them to our work in the discussion (lines 517-526) section. We sincerely hope the related works are now extensive enough to provide sufficient backdrop for our work.

Reviewer 2 Report

Comments and Suggestions for Authors

Thank you for addressing my comments. Most of my concerns have been well justified. However, after reading other reviewers’ comments, and the reviewed text, the novelty and contributions are not clear in the manuscript.

If I understand correctly, there are two main contributions: The proposal of a neural network for classifying pre- and post- rehabilitation states and the assessment of most predictive features (without taking into account interdependencies). This should be clearer in the abstract.

Subsection 1.2 title should be: Contributions and article organization.

Subsection 3.2 should be renamed (and also line 385), since the authors only have one ML model and different connectivity methods. I understand that in this subsection, the authors are assessing the performance of these methods through the same ML model.

Confusion matrices are too small and numbers cannot be clearly written. Enlarge all the related text, please.

Author Response

Response to Reviewer 2’s comments

Yes

Can be improved

Must be improved

Not applicable

Does the introduction provide sufficient background and include all relevant references?

(x)

( )

( )

( )

Are all the cited references relevant to the research?

(x)

( )

( )

( )

Is the research design appropriate?

(x)

( )

( )

( )

Are the methods adequately described?

( )

(x)

( )

( )

Are the results clearly presented?

( )

(x)

( )

( )

Are the conclusions supported by the results?

(x)

( )

( )

( )

Comments and Suggestions for Authors:

We thank the reviewer for their patience and comments to help us improve our paper. The point-by-point responses to the comments are provided below, while changes made to the body of the text are marked in red therein.

Thank you for addressing my comments. Most of my concerns have been well justified. However, after reading other reviewers’ comments, and the reviewed text, the novelty and contributions are not clear in the manuscript.

We thank the reviewer for their comments on our manuscript. We have taken additional care to highlight the contributions of our study in the paper. First, the last sentence of the abstract was re-written to better reflect the main contributions of our paper (lines 14-17). Second, the last paragraph of the “Related works” section was moved to the first paragraph of the now-renamed “Contributions and article organization” section (lines 141-145). In accordance with the comments of another reviewer, more related works were added, but to highlight our contributions and the novelty of our work in comparison with these, an explicit statement of novelty was added (lines 102-145). We have also added a comparison with previous works to the discussion section (lines 517-526), that highlights the novelty of our approach.

If I understand correctly, there are two main contributions: The proposal of a neural network for classifying pre- and post- rehabilitation states and the assessment of most predictive features (without taking into account interdependencies). This should be clearer in the abstract.

We thank the reviewer for suggesting the changes to improve the clarity of our abstract. To improve the abstract, the last sentence was re-written and expanded upon, to more clearly present our contributions and the novelty of our approach (lines 14-17). We see the main contributions of our work in providing a clear methodology of utilizing a relatively small amount of data to achieve valid and meaningful results with modern machine learning approaches, and in systematically evaluating and comparing the utility of often-used connectivity metrics as input features. We demonstrate the proposed methodology to classify pre- and post-rehabilitation states, where we also assess the predictivity of different feature types. However, in this work, we do not fully investigate which are the most predictive features. The selection of the 10% most significant features is important only due to the reduction of a search space, which as such enables application of less complex classification models. With this approach we still keep all the significant features while we do not further asses their importance. Their importance and interdependence is implicitly taken into account by the neural network. We agree, and we mention it in the paper, that there is an evident step for future research - to investigate the predictivity of features, which can be performed by neural network explainability approaches.

Subsection 1.2 title should be: Contributions and article organization.

We changed the subsection title in accordance with Your suggestion (line 140).

Subsection 3.2 should be renamed (and also line 385), since the authors only have one ML model and different connectivity methods. I understand that in this subsection, the authors are assessing the performance of these methods through the same ML model.

We thank the reviewer for pointing out that the subsection title is unclear. To improve the clarity of our manuscript, we have renamed this section to “Evaluation of ML models for the best-performing methods”, accordingly (line 419). As we also detail in the paper (lines 362-393), we trained a model for each connectivity metric and frequency band combination, that is, on the 10% of the electrode pair connectivities that exhibited the most statistically significant changes after rehabilitation. One model each was trained for PLV in the delta band, PLV in the theta band, PLV low gamma, PLV high gamma, wPLI delta, wPLI theta, … etc., for each combination of connectivity metric and frequency band studied. The exception was the Granger Causality, which was undivided by frequency. While the architecture of the models was the same in every case, meaning they were three-layered fully connected neural networks, with 10, 5 and 2 nodes in each layer, and RELU activation functions between them. Furthermore, we consider the entire system, from feature extraction to model training and output generation as a single model. Keeping this in mind, we are reluctant to name our systems as one single model, as they are trained on different data and differ greatly in predictive capacity.

Confusion matrices are too small and numbers cannot be clearly written. Enlarge all the related text, please.

We thank the reviewer for bringing this issue to our attention. The matrices were re-drawn and all font sizes increased for improved readability (Figure 7). We hope these alterations are satisfactory, and the figures are now clearer.

Reviewer 3 Report

Comments and Suggestions for Authors

The Authors addressed my observations and the paper improved.

Author Response

Response to Reviewer 3’s comments

Yes

Can be improved

Must be improved

Not applicable

Does the introduction provide sufficient background and include all relevant references?

(x)

( )

( )

( )

Are all the cited references relevant to the research?

(x)

( )

( )

( )

Is the research design appropriate?

(x)

( )

( )

( )

Are the methods adequately described?

(x)

( )

( )

( )

Are the results clearly presented?

(x)

( )

( )

( )

Are the conclusions supported by the results?

(x)

( )

( )

( )

Comments and Suggestions for Authors:

The Authors addressed my observations and the paper improved.

We sincerely thank the reviewer for their previous comments and assistance with improving our manuscript.